# Selected Properties of Bio-Based Layered Hybrid Composites with Biopolymer Blends for Structural Applications

**DOI:** 10.3390/polym14204393

**Published:** 2022-10-18

**Authors:** Aneta Gumowska, Eduardo Robles, Arsene Bikoro, Anita Wronka, Grzegorz Kowaluk

**Affiliations:** 1Department of Technology and Entrepreneurship in Wood Industry, Warsaw University of Life Sciences, Nowoursynowska Str. 159, 02-776 Warsaw, Poland; 2University of Pau and the Adour Region, E2S UPPA, CNRS, Institute of Analytical and Physicochemical Sciences for the Environment and Materials (IPREM-UMR 5254), 403 Rue de Saint Pierre, 40004 Mont de Marsan, France

**Keywords:** composite, mechanical properties, blends, polylactide, polycaprolactone, polyhydroxybutyrate, microcrystalline cellulose, triethyl citrate

## Abstract

In this study, layered composites were produced with different biopolymer adhesive layers, including biopolymer polylactic acid (PLA), polycaprolactone (PCL), and biopolymer blends of PLA + polyhydroxybutyrate (PHB) (75:25 *w*/*w* ratio) with the addition of 25, 50% microcrystalline cellulose (MCC) and 3% triethyl Citrate (TEC) for these blends, which acted as binders and co-created the five layers in the elaborated composites. Modulus of rupture (MOR), modulus of elasticity (MOE), internal bonding strength (IB), density profile, differential scanning calorimetry (DSC), thermogravimetric analysis (TGA), and scanning electron microscopy (SEM) analysis were obtained. The results showed that among the composites in which two pure biopolymers were used, PLA obtained the best results, while among the produced blends, PLA + PHB, PLA + PHB + 25MCC, and PLA + PHB + 25MCC + 3TEC performed best. The mechanical properties of the composites decreased with increases in the MCC content in blends. Therefore, adding 3% TEC improved the properties of composites made of PLA + PHB + MCC blends.

## 1. Introduction

The continuous development of science and technology increases the demand for environmentally friendly products of natural origin and the increased reuse of forestry and agricultural byproducts that are treated mostly as waste [1,2]. Due to concerns related to the depletion of petroleum and greenhouse gas emissions resulting from the production of petroleum products, the use of renewable, recyclable, and compostable raw materials is becoming increasingly desirable [3]. In recent years, environmental regulations have forced producers of wood-based composites to think about using sustainable materials in producing new products. This is one of the reasons why the use of alternative raw materials in wood-based composites is of growing interest [4,5]. New political strategies aim to improve efficiency and reduce impacts on health and the environment, and are also an essential element in promoting competitiveness and the idea of sustainable development [6]. Trends in wood-based products for commodities show an increase in demand, which corresponds to a combination of different factors such as aesthetic aspects, awareness of the origin of products, service life, and performance. However, some of these factors can be contradictory, as products from natural origin have a shorter service life than petroleum products, and products with enhanced performance usually imply a more complex disposal of the residues. On the other hand, there is pressure for new products to perform perfectly during use without harming health and the environment, and, at the end of their life cycle, to be easily reduced, reused, recycled, composted, or recovered as energy. The elaborate production of biocomposites, defined as materials composed of at least two distinct constituent materials (with at least one being naturally derived), using biopolymers and natural fibers is the answer to the current market needs. Unlike petroleum products, natural products can be entirely recovered for further applications [7,8].

Currently, adhesives based on petrochemical products contain volatile organic compounds (VOCs), and toxic chemicals are commonly used in wood-based materials, particleboards, fiberboards, and layered panels [9]. Synthetic adhesives dominate the production process for wood-based materials due to their low manufacturing cost, superior bonding properties and board reliability [10]. However, adhesives based on formaldehyde come from non-renewable petrochemical sources. Despite many efforts to develop low-formaldehyde emission binders, as well as the use of formaldehyde scavengers [11], these have the disadvantage of free formaldehyde emissions during manufacturing and during the use of the final products, which have a detrimental effect on health and the environment [12,13]. In addition, urea-formaldehyde (UF) and other amine resins are a severe concern for human health due to their pathogenic character, which has been extensively studied and demonstrated. Factors driving the development of renewably sourced adhesives as a promising alternative to synthetic adhesives are the regulation of formaldehyde emissions from wood-based industries and market interest in bio-adhesives [14]. Soubam et al. [10] produced plywood with biopolymers as an adhesive; in this case, elaboration and tests of layered composites were performed based on two biopolymers: natural rubber latex (NRL) and dimethylol dihydroxy ethylene urea (DMDHEU) cross-linked with rice starch. Researchers proved that synthetic-based adhesives could be sustainably replaced with selected blends of biopolymers. Based on the results, the mechanical properties of plywood bonded with 10 g NRL + 10 g cross-linked rice starch DMDHEU showed MOR, MOE, and IB values that met the standards.

The use of polymer blends to bond fibers for wood-based products seems an attractive substituent because of the low cost and suitability for a wide range of applications. In this sense, polymer blends with either lignocellulosic fibers or wood flour have been studied for several years. However, the current polymer consumption per human has provoked increasing concern regarding waste management and environmental impact. Post-service-life processes associated with plastics, such as recycling, reducing, and reusing, have become necessary and commendatory, and several laws have been made for this purpose worldwide. Therefore, the use of polymers from renewable materials that represent less harm to the environment has increased in research and industry. There is a steadily growing interest in developing non-toxic and environmentally friendly adhesives that have adhesion properties similar to synthetic commercial resins. The most common biodegradable synthetic polymers are aliphatic polyesters such as polylactic acid (PLA), polyglycolic acids (PGA), polycaprolactone (PCL), and polyhydroxyalkanoates (PHA), among others. Above all, PLA is widely used in commodities for many applications, including grocery and composting bags, automobile panels, textiles, and bio-absorbable medical materials [15,16,17]. The thermal properties of this material make it an attractive option for thermoplastic processes such as extrusion, injecting molding, blow molding, thermoforming, sheet forming, and film forming [18,19]. PLA is a semi-crystalline aliphatic polyester with a low glass-transition temperature and high transparency. On the other hand, it can be highly brittle [20] and has low heat resistance [21] and poor barrier properties [22], although these properties can be improved with the addition of polymers with higher crystallinity. PLA may not biodegrade at room temperature, but it does biodegrade in controlled composting systems [23]. Polycaprolactone (PCL) is widely used in applications including biomedical materials [24], hot melt adhesives [25], and degradable plastics [26]. The advantages include its excellent biodegradability, high flexibility, biocompatibility, and processibility [27].

PHB is one of the most used polyhydroxyalkanoates; it can be synthesized by controlled bacterial fermentation [28], which has attracted increasing attention. PHB presents high crystallinity [29]; nevertheless, its main drawback is its melting temperature (Tm) which is approximately 170–180 °C, close to its degradation temperature of approximately 270 °C, yielding a small processing range for melt extrusion [30]. Blending two or more polymers with different properties to produce composite materials is a well-known strategy to obtain specific physical properties without the need for complex polymeric systems [31]. PLA and PHB blends have been intensively studied because of the good synergy they can form, with PHB increasing the crystallinity of the blend [32,33,34] and PLA increasing the stiffness [35]. The mechanical blending of these two polymers can be achieved in a melt state because of their similar melting temperatures, which allows for better blending. Many studies concluded that the best blend is achieved by blending 75 wt% PLA with 25 wt% PHB [32,35,36,37].

Polymers reinforced with natural fibers are replacing synthetic fiber-reinforced plastics in different industrial sectors, including the automotive industry, packaging, and furniture production, providing lighter materials with better thermal properties [38]. This investigation aimed to assess the impact of natural biopolymer binders on selected mechanical and physical properties of a five-layer lignocellulosic composite produced with different biopolymer layers (including, for example, PLA and PCL), using biopolymer blends as an adhesive and to co-create the five layers in the composites. In the light of the state of art mentioned above, this study fills the gap between the development of formaldehyde-free binders for wood, made of renewable resources, and that of wood-based layered composites modified by additional layers made of biopolymers and their blends.

## 2. Materials and Methods

### 2.1. Materials

This study produced layered composites from beech (*Fagus sylvatica* L.) veneers. The nominal dimensions of the commercial veneers were 2500 mm × 200 mm × 0.60 mm, length × width × thickness, respectively. The veneers were cut into 200 mm × 200 mm sheets. The moisture content of ca. 5% of every veneer was measured using an ultrasonic moisture control device. Pure, laboratory-purpose polylactide (PLA, Sigma-Aldrich, product no. 38534, Burlington, MA, USA), polycaprolactone (PCL, Sigma-Aldrich, product no. 704105) in drops with a diameter of 3 mm, and five variants of biopolymer blends, obtained under laboratory conditions, were used as binders. The following components were used to achieve biopolymer blends: PLA was provided by Futerro (Belgium) at extrusion grade; Polyhydroxybutyrate (PHB, P309E) was provided by Biomer (Germany); Sigma-Aldrich provided microcrystalline cellulose (MCC); triethyl citrate (TEC) was provided by Acros Organics.

### 2.2. Biopolymer Blend Elaboration

The PLA–PHB masterbatch (MB) was blended in a 75:25 *w*/*w* ratio according to recommendations from the literature [29,33], and composites were elaborated by mixing MB with different contents of MCC and TEC, as shown in Table 1. Blends were manufactured using a twin-screw extruder (M250, LabTech Engineering, Thailand) with a screw speed of 30–200–100–100 rpm (feed–mix–extrusion) and a temperature profile process of 180–185–190–195 °C. Composites were extruded twice to guarantee dispersion and then granulated into pellets.

### 2.3. Manufacturing of Biopolymer Adhesive Layers

The biopolymer adhesive layers were manufactured with an intended thickness of 1 mm. The granules were manually spread onto a frame mold with an average total of 62 g (1550 g m^−2^) over PTFE mats and placed on pressing steel plates. The granules were evenly distributed over the entire surface, which was limited by the frame (Figure 1a). The first stage involved heating slightly above the melting point of the binder; therefore, only the bottom steel plate with biopolymers/blends spread on the PTFE mat was placed in the press. Adequately thick spacer bars allowed the press shelves to be closed without the upper shelf contacting the granules, while simultaneously trying to provide heat from both sides. This treatment with the bottom steel plate and spacer bars made it possible to control the melting of these materials. The heating time was 10 min, enough to reach the melting point of the blends (Figure 1b). When the biopolymers and blends achieved a molten consistency, they were covered from above with a second PTFE mat and a steel plate and re-inserted between the press shelves to be pressed to a thickness of 1 mm. The temperature of the press was 185 °C. A water bath was used to cool the layers after removal from the press, and then the obtained adhesive layers were cut along the sides of the frame to obtain an adhesive sheet (Figure 1c).

### 2.4. Layered Composite Manufacturing

Five-layer composites (alternating layers of veneer with biopolymer layers) were manufactured with dimensions 200 mm × 200 mm with an average thickness of 3 mm. The middle layer grain directions were oriented 90° relative to the surface veneers. As a result, seven types of composites were produced with different adhesive layers (subsequently denoted by the shortcodes listed in Table 2), and no less than four layered panels of each binder type. The composites were pressed for 5 min without pressure, after which they were overheated and the adhesive layers were allowed to melt between the veneers; then, 0.6 MPa pressure was applied for 1 min, followed by 1.5 MPa for 1 min. The total pressing time was 7 min. The press temperature was 185 °C. Following the research plan, the produced composites were conditioned in ambient conditions (20 °C; 65% R.H.) to a constant weight for seven days before being cut.

### 2.5. Mechanical and Physical Properties

The physical and mechanical properties were tested according to European standards. The moduli of rupture (MOR) and moduli of elasticity (MOE) were determined according to EN 310 [39] and were reported as the average of ten measurements. Internal bond (IB) was measured according to EN 319 [40]. Five samples of the layered lignocellulosic composite for each binder variant were used for bond quality. All mechanical properties were examined with an INSTRON 3369 (Instron, Norwood, MA, USA) standard laboratory testing machine. The density profile (DP) of samples was analyzed using a DA-X measuring instrument (GreCon, Alfeld, Germany). Measurement based on direct scanning X-ray densitometry was carried out with a speed of 0.05 mm s^−1^ across the panel thickness with a sampling step of 0.02 mm. The nominal dimensions of all samples were 50 mm × 50 mm. Graphs of the density distribution for each composite type were obtained based on three replicates. Differential Scanning Calorimetry (DSC) tests were performed using a DSC Q20 Instrument (TA Instruments, New Castle, DE, USA). The measurements were carried out at a heating rate of 10 °C min^−1^ under an inert gas (nitrogen) atmosphere with a flow rate of 50 mL min^−1^. Samples of 5 mg were tested with two repetitions. Thermogravimetric analysis (TGA) was performed on a Q500 (TA Instruments, New Castle, DE, USA) apparatus in air (flow rate 40 mL min^−1^) in a temperature range of 50–600 °C at a heating rate of 10 °C min^−1^. Samples of 5–10 mg were tested with two repetitions. Scanning electron microscopy images (SEM) were obtained with a Quanta 200 (FEI, Hillsboro, OH, USA) scanning electron microscope. Pictures of the cross-sections of the manufactured five-layer composites were obtained with a NIKON SMZ 1500 (Kabushiki-gaisha Nikon, Minato, Tokyo, Japan) optical microscope.

### 2.6. Statistical Analysis

Analysis of variance (ANOVA) and *t*-test calculations were used to test (α = 0.05) for significant differences between factors and levels, where appropriate, using IBM SPSS statistical software (IBM, SPSS 20, Armonk, NY, USA). In addition, a comparison of the means was performed by employing the Duncan test when the ANOVA indicated a significant difference.

## 3. Results and Discussion

TGA and DSC analysis was carried out for more exhaustive characterization of all produced biopolymer adhesive layers (Figure 2 and Figure 3). The obtained TGA results give information regarding the thermal resistance of the tested materials. The results for samples of PLA and PCL were widely analyzed and compared to the findings in Gumowska et al. (2021) [41]. TGA analysis recorded the degradation temperature, which should not be exceeded during further tests with these materials. TGA analysis of the biopolymers and blends showed one main thermal degradation stage at temperatures of 280–430 °C, with a mass loss of approximately 90%. Figure 2 presents two characteristic curve profiles. The first is a smooth transition (PLA and PCL) and the second shows a deflection characteristic of materials consisting of two or more components (biopolymer blends). The data in Table 3 display the thermal stability established for 50% and 80% weight loss of the tested samples. The higher temperature obtained is related to higher thermal resistance. The highest thermal stability was noted for PCL. The rest of the samples showed similar thermal stability between them. The endothermal melting process of PLA and PCL was similar to that found in published data. In Figure 3 multiple melting peaks can be seen, which were previously reported for PHB and copolymers. These multiple peaks could be caused by melting, recrystallization, and remelting during heating; polymorphism; different molecular weight species; different lamellar thickness, perfection, or stability; and physical aging or relaxation of the rigid amorphous fraction, among others [42]. For example, melting–recrystallization–remelting was considered the cause of complex double melting in PHBV [43]. A similar mechanism may be supposed for the PHB reference, but the occurrence of different molecular weight species due to chain scission during melt processing or the presence of β crystals in addition to the common α form of PHB should not be excluded [44].

The results of the density profiles are summarized in Figure 4. The average densities of all composites ranged from 980 to 1080 kg m^−3^. The density profiles for individual samples were symmetrical to the middle of the thickness of the composites; therefore, the graph presents the density profiles for their axis of symmetry to facilitate analysis. The graph shows the estimated boundaries of the layers in the composites. The tested composites consisted of five layers, alternating between veneer layers and biopolymer layers. Regardless of the biopolymer or blend layer used, the shapes of the profiles did not differ significantly. The most remarkable information corresponds to PCL, which is usually less dense than PLA and its derived blends. Every determined profile shows an increase in density characteristic of the layer materials, precisely in the bonding zones, which is related to compaction of the resin due to resistance of the wood to impregnation [45,46]. This image justifies calling the produced composites multi-layered composites, as they could be perceived as five-layer panels, of which the biopolymer layer acted as a binder, and separate layers were visible even to the naked eye. The bonding lines are accurately shown on the graphs; they were flat over the entire section of each sample’s width. The highest average values of bonding line density were recorded for biopolymer blends with 50% MCC due to the density of cellulose (around 1500 kg m^−3^). Another remarkable fact is the uneven distribution of densities at the 90° veneer (right side of the plots), which corresponds to an inverted section in comparison to the face veneer, thus presenting a different structure along the section with a lower density in the zones with tracheids.

The average values of modulus of rupture (MOR) and modulus of elasticity (MOE) under three-point bending stress for the tested composites are presented in Figure 5. The highest average value of MOR (153 N mm^−2^) was found for PLA + PHB samples, while the lowest (93 N mm^−2^) was found for PLA + PHB + 50MCC. In the case of MOE, the highest average value was achieved for PLA + PHB (13,718 N mm^−2^) and MOR. The lowest was for PCL (11,437 N mm^−2^). Adding MCC to PLA + PHB reduced MOR and MOE, while adding 3% TEC increased MOR and MOE in composites with 25 and 50% MCC in biopolymer blends. The tasks of plasticizers are, among others, to enhance polymer chain mobility [47] and to reduce intermolecular interactions, thus giving the material greater flexibility and a plasticization effect [48]. The addition of 3% TEC to blends with 25% and 50% MCC caused the bonding to anchor deeper in the wood structure than blends without added TEC, which was confirmed by the density profile graph for these samples and potentially also affected the mechanical properties [16]. Based on statistical analysis, there were no statistically significant differences between the average MOR values for PLA, PLA + PHB, PLA + PHB + 25MCC, PLA + PHB + 25MCC + 3TEC, PLA + PHB + 50MCC + 3TEC, or between PCL and PLA + PHB + 50MCC binders.

The results of internal bonding (IB) tests are presented in Figure 6. The outcomes show that the highest average value of IB was that of PLA (8.67 N mm^−2^), and the lowest value was found for PLA + PHB + 50MCC (3.29 N mm^−2^). When analyzing the biopolymer blends, it was noticed that composites with 25% MCC fared better than 50% MCC. Increasing to 25% and 50% MCC in blends resulted in decreases in internal bonding of more than 4% and 37%, respectively. This occurred because the mixture of PLA, PHB, and cellulose resulted in packed phases rather than a new polymer or copolymer, potentially generating contact zones prone to failure due to bad mixing. Even if the blends with PLA and PHB showed good properties overall, bad blending at the extruder or a shorter pre-melting time during the first steps of pressing might generate interstices in which fracture could occur. This was further noted in samples containing MCC as the powder also generated surface–surface defects, as shown before [18].

On the other hand, adding 3% TEC to the blends with 25% MCC increased IB by 14%, whereas adding 3% TEC to 50% cellulose blends increased IB by 28%, thus proving the positive contribution of TEC to composite blends. Based on statistical analysis, statistically-significant differences existed between the average values of IB for PLA and the rest of the samples. There were no statistically significant differences between PCL, PLA + PHB + 50MCC, PLA + PHB + 50MCC + 3TEC, and PLA + PHB, PLA + PHB + 25MCC, and PLA + PHB + 25MCC + 3TEC. The two major forms of damage to the samples resulting from the IB test are presented in Figure 7. The first type of damage occurred in the near-surface zone, along with partial destruction in the wood structure obtained as a biopolymer adhesive. The second group includes samples in which the damage took place in the near-surface zone along with destruction on the surface of the adhesive layers.

SEM analysis and optical microscopy observation showed the penetration of the biopolymers and biopolymer blends into the wood structure (Figure 8.). There was a recognizable difference in the adhesion zone between the adhesive layers and the wood, where partial penetration of the binder into the pores of the wood could be seen. Penetration into the wood structure is visible in the optical microscope photographs; for an example, see the images of PLA + PHB + 25MCC and the same binder with the addition of 3% TEC (Figure 8e,f). It can be seen that the veneers were not 100% impregnated with the binder, which was also confirmed by the density profiles of the manufactured five-layer composites and which allows for the presence of an exposed face of the veneer with no traces of any polymer resin, which in turn permits either a raw or a finished appearance on the surface face of the elaborated composites. In addition, the addition of TEC affected the penetration depth into the pores of the wood, which could be the origin of the better mechanical properties shown by these composites, in which the wood structure acted as the skeleton of the solidified polymer resin, resulting in a tougher composite.

## 4. Conclusions

In the above study, five-layer composites were produced in which biopolymer adhesive layers (PLA, PCL, blends of PLA + PHB (75:25 *w*/*w* ratio) with or without the addition of 25% or 50% MCC, and 3% TEC for these blends) were used as binders. The density profiles of the composites manufactured with biopolymers and biopolymer blends as binders did not deviate from the characteristic profiles of plywood, with increased densities at the bonding line. The produced layers acted as binders and co-created the five layers in the composites. The mechanical properties of the composites decreased with increases in the amount of MCC in blends. MOR, MOE, and IB values for PLA + PHB + 25MCC and PLA + PHB + 50MCC decreased by 11% and 39%, 6% and 15%, and 4% and 40%, respectively. Therefore, adding TEC improved the properties of composites made of PLA + PHB + MCC blends. Among the composites in which two pure biopolymers were used, PLA obtained the best results, while among the produced blends, PLA + PHB, PLA + PHB + 25MCC, and PLA + PHB + 25MCC + 3TEC performed best.

The results achieved herein, regarding an attempt to produce layered wood-based composites with different biopolymers and their blends as special properties layers and binders, allow for the conclusion that it is possible to create a formaldehyde-free wood-based layered composite that enhances the properties of both materials—wood and biopolymer. However, additional work should be performed in the field of biopolymer blend composition to improve adhesion to wood.

## Figures and Tables

**Figure 1 polymers-14-04393-f001:**
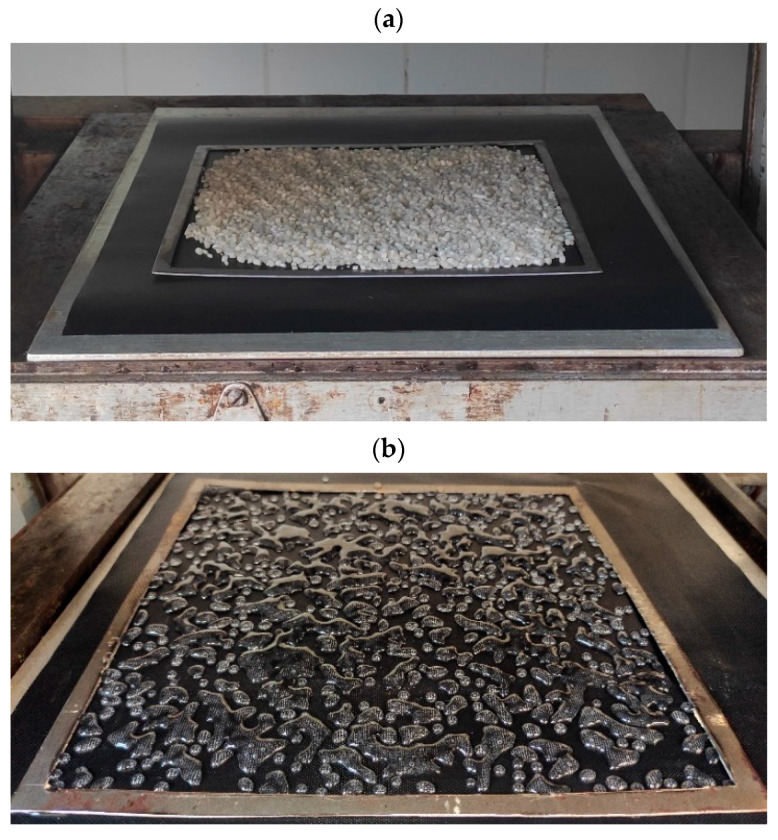
(**a**) Overheating phase of the granules of biopolymers and biopolymer blends; (**b**) molten pellets (PLA in this case); (**c**) adhesive biopolymer layer.

**Figure 2 polymers-14-04393-f002:**
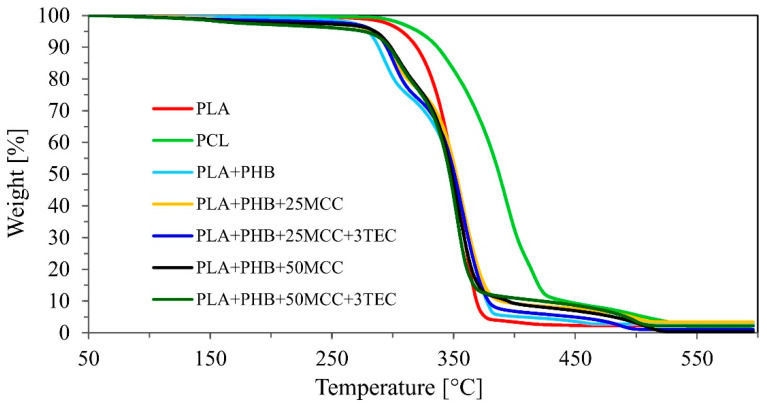
Thermogravimetric analysis (TGA).

**Figure 3 polymers-14-04393-f003:**
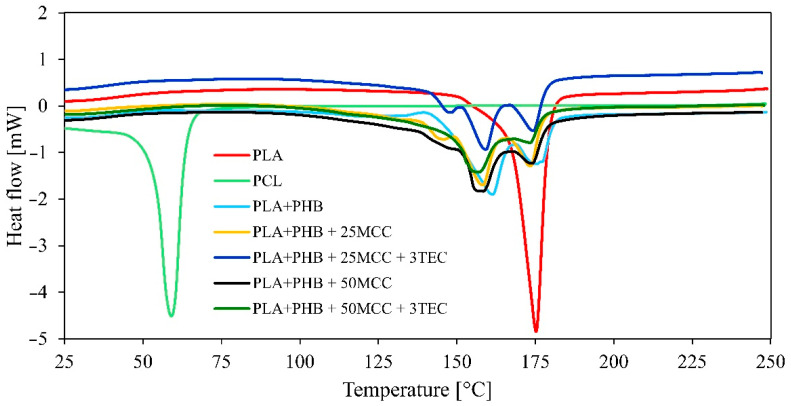
Differential scanning calorimetry (DSC).

**Figure 4 polymers-14-04393-f004:**
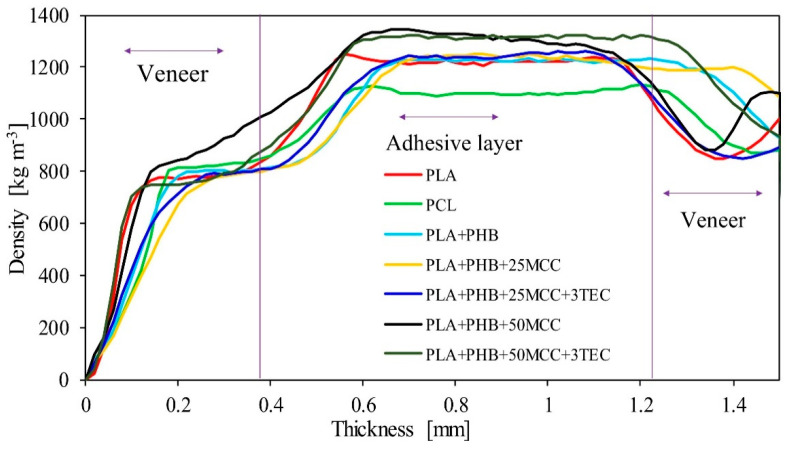
Density profiles of the produced five-layer composites.

**Figure 5 polymers-14-04393-f005:**
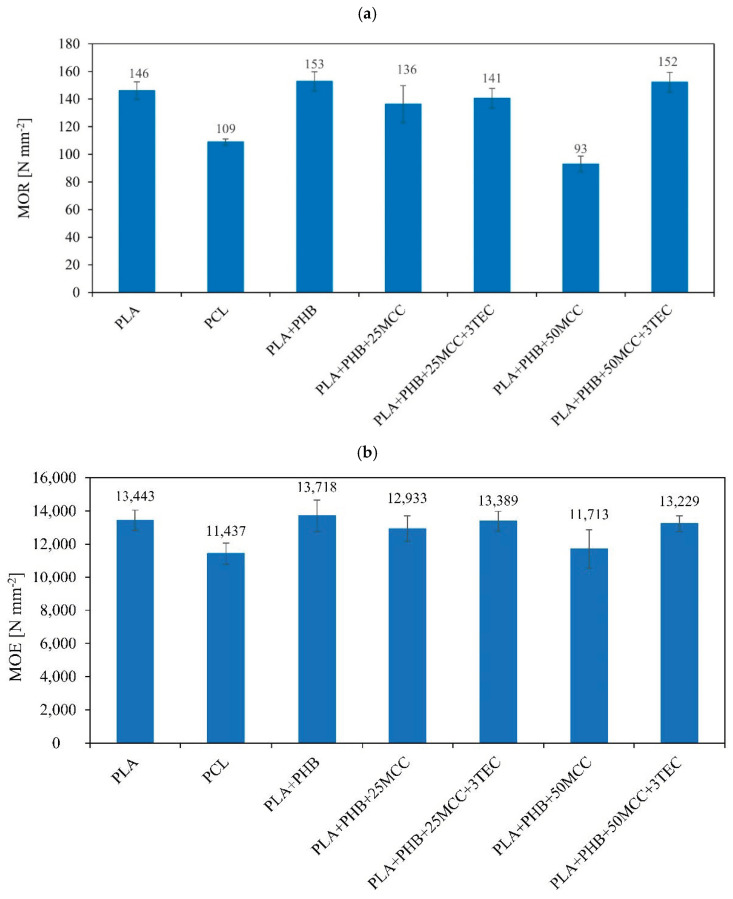
(**a**) Modulus of rupture (MOR) and (**b**) modulus of elasticity (MOE) of the tested five-layer composites.

**Figure 6 polymers-14-04393-f006:**
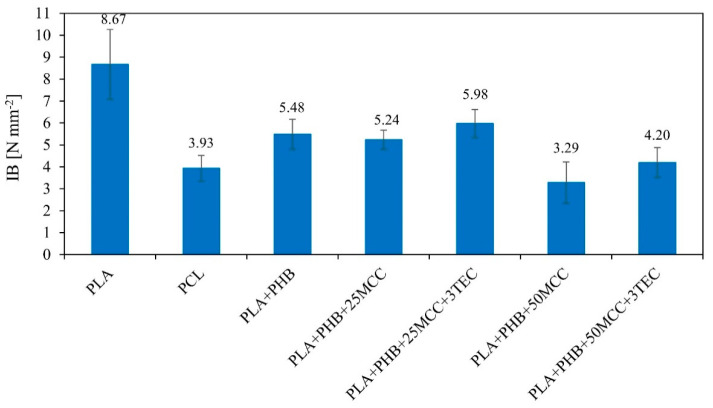
Internal bond (IB) results of tested five-layer composites.

**Figure 7 polymers-14-04393-f007:**
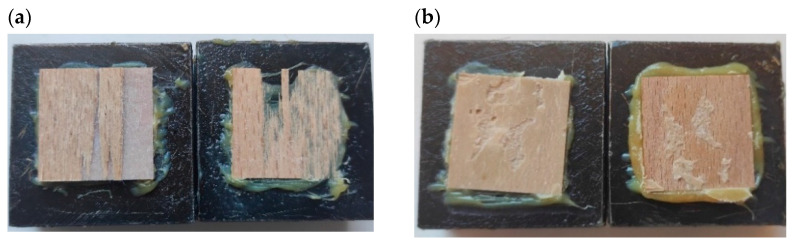
The two major forms of damage after the IB test: (**a**) near-surface zone along with partial destruction in the structure of the wood; (**b**) near-surface zone along with destruction in the adhesive layers.

**Figure 8 polymers-14-04393-f008:**
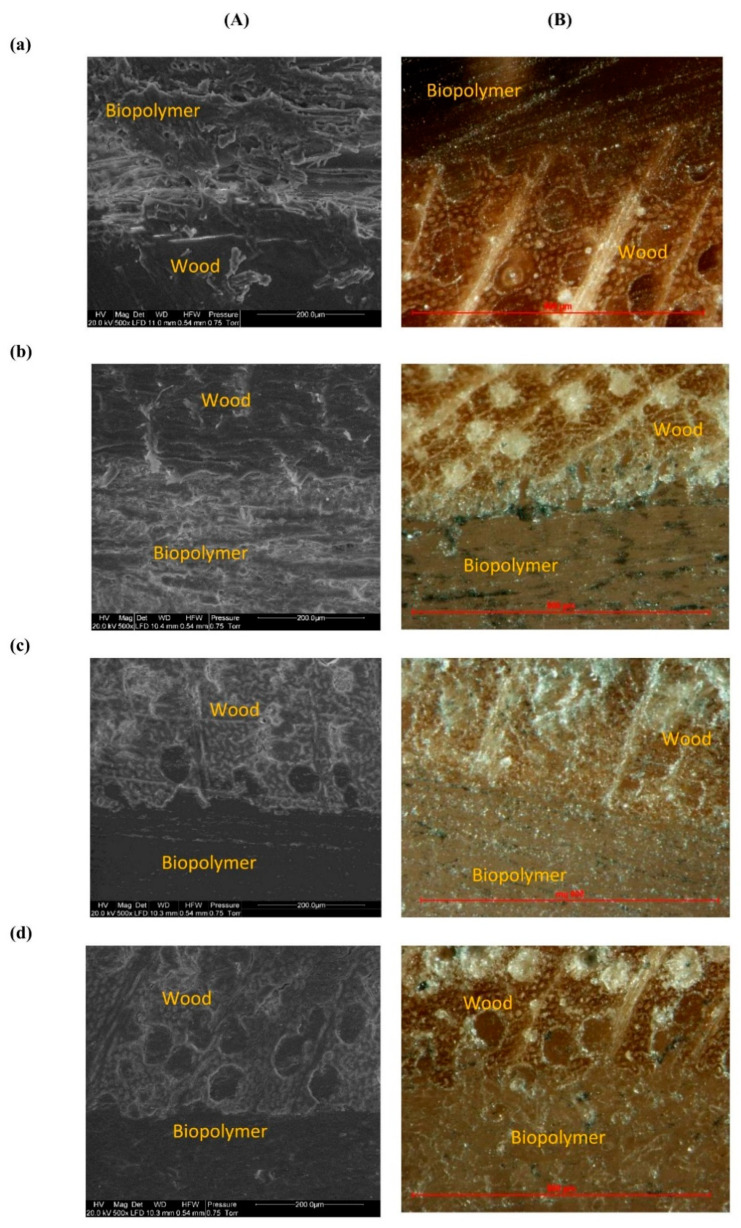
(**A**) SEM pictures (×500) and (**B**) optical microscope photos of the cross-section of composites (10×): (**a**) PLA, (**b**) PCL, (**c**) PLA + PHB, (**d**) PLA + PHB + 25MCC, (**e**) PLA + PHB + 25MCC + 3TEC, (**f**) PLA + PHB + 50MCC, (**g**) PLA + PHB + 50MCC + 3TEC. The wood and binder layers are indicated in the pictures.

**Table 1 polymers-14-04393-t001:** Composition and shortcode for every elaborated biopolymer blend.

Matrix	PLA [%]	PHB [%]	MCC [%]	TEC [%]
PLA + PHB (75:25)	75.0000	25.0000	0.0000	0.0000
PLA + PHB (75:25)	54.5625	18.1875	24.2500	0.0000
PLA + PHB (75:25)	55.5000	18.5000	25.0000	3.0000
PLA + PHB (75:25)	37.5000	12.5000	50.0000	0.0000
PLA + PHB (75:25)	36.3750	12.1250	48.5000	3.0000

**Table 2 polymers-14-04393-t002:** Shortcodes for every elaborated composite.

Biopolymer Layers	Sample
Polylactide	PLA
Polycaprolactone	PCL
PLA + PHB	PLA + PHB
PLA + PHB + 25% microcrystalline cellulose (MCC)	PLA + PHB + 25MCC
PLA + PHB + 25% MCC + 3% triethyl citrate (TEC)	PLA + PHB + 25MCC + 3TEC
PLA + PHB + 50% MCC	PLA + PHB + 50MCC
PLA + PHB + 50% MCC + 3% TEC	PLA + PHB + 50MCC + 3TEC

**Table 3 polymers-14-04393-t003:** Thermal stability estimators for the investigated biopolymers and biopolymer blend samples.

Tested Materials	Mass Loss	
	50%	80%
	°C	
PLA	349.3	363.3
PCL	387.4	413.8
MB	350.3	369.7
MB + C1	352.2	370.5
MB + C2	350.6	368.3
MB + C3	348.5	364.9
MB + C4	346.8	361.0

## Data Availability

The data presented in this study are available on request from the corresponding author.

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
