# Peer review of "Selected Properties of Bio-Based Layered Hybrid Composites with Biopolymer Blends for Structural Applications"

_polymers, 2022, doi:10.3390/polym14204393_

Round 1
Reviewer 1 Report
Overall, in my opinion, the manuscript is of very high quality, and the research is up-to-date and thorough. Therefore, I have only a few recommendations for the esteemed authors.
In the "Introduction", it would be good to insert a part about the necessity and relevance of the research.
Please consider reformatting the sentence "Different mechanical and physical properties were carried out, namely modulus of rupture (MOR), modulus of elasticity (MOE), internal bonding strength (IB), as well as density profile, DSC, TGA, and SEM analysis." (lines 16-18). For example, instead of "Different", you can use "Main", "Standardized", "Exploitation", or "The mechanical and physical properties were carried out……" or "Modulus of rupture (MOR), modulus of elasticity ( MOE), internal bonding strength (IB), as well as density profile, DSC, TGA, and SEM analysis were carried out".
The Introduction is well prepared, placing the present study in the context of the search for eco-friendly solutions for the sustainable production of biopolymer composites. Here, I think it could be supplemented with short data and analysis of natural bio-based alternatives, like lignin and tannin, to fossils binders.
The "Materials and Methods" and "Results" parts are very well prepared. Here I have only one recommendation - I ask the respected authors to consider whether point three should not only be "Results" but "Results and Discussion" or "Results and Analyses", for example.
Since the authors have done a significant amount and quality of work, this should be better highlighted in "Conclusions". I ask that this part be expanded with the study's main novelty and emphasize the main results achieved.
The references cited are appropriate.
Author Response
Overall, in my opinion, the manuscript is of very high quality, and the research is up-to-date and thorough. Therefore, I have only a few recommendations for the esteemed authors.
In the "Introduction", it would be good to insert a part about the necessity and relevance of the research.
The comment regarding research relevance has been added to the “Introduction”.
Please consider reformatting the sentence "Different mechanical and physical properties were carried out, namely modulus of rupture (MOR), modulus of elasticity (MOE), internal bonding strength (IB), as well as density profile, DSC, TGA, and SEM analysis." (lines 16-18). For example, instead of "Different", you can use "Main", "Standardized", "Exploitation", or "The mechanical and physical properties were carried out……" or "Modulus of rupture (MOR), modulus of elasticity ( MOE), internal bonding strength (IB), as well as density profile, DSC, TGA, and SEM analysis were carried out".
The paragraph including mentioned sentence has been modified.
The Introduction is well prepared, placing the present study in the context of the search for eco-friendly solutions for the sustainable production of biopolymer composites. Here, I think it could be supplemented with short data and analysis of natural bio-based alternatives, like lignin and tannin, to fossils binders.
The mentioned alternatives have been added in additional citations.
The "Materials and Methods" and "Results" parts are very well prepared. Here I have only one recommendation - I ask the respected authors to consider whether point three should not only be "Results" but "Results and Discussion" or "Results and Analyses", for example.
The title of the paragraph has been changed.
Since the authors have done a significant amount and quality of work, this should be better highlighted in "Conclusions". I ask that this part be expanded with the study's main novelty and emphasize the main results achieved.
A sufficient comment has been added.
The references cited are appropriate.
Thank you!
Please find attached file

Reviewer 2 Report
The manuscript is well-prepared and can be published after minor revisions.Please consider simplifying the title, as it is unnecessary long.
Please explain abbreviations during first use (PLA, PCL, PHB, MCC, TEC, DSC, TGA, etc.). PLA defined for example in line 78 first (used in the abstract).
Lines 27-29: There are many review articles about the use of agricultural (and wood) waste for composites, authors should mention some of the newest.
Lines 44-47: Authors should define what they mean by biocomposites.
Lines 48-50: Please add also some statistics.
Lines 54-59: Not only UF, but also PF, MF, MUF, etc.
Please discuss more deeply in the Introduction previous research into physical and mechanical properties of bio-based layered hybrid composites.
The relevance should be mentioned in the Introduction.
The results and discussion part are well written.
More discussion with previous research is needed in the discussion part.
Please add also limitations of your research, novelty and implicationsfor further research and industry.
Author Response
The manuscript is well-prepared and can be published after minor revisions.
Please consider simplifying the title, as it is unnecessary long.
The title has been simplified.
Please explain abbreviations during first use (PLA, PCL, PHB, MCC, TEC, DSC, TGA, etc.). PLA defined for example in line 78 first (used in the abstract).
The abbreviations have been explained during the first use (in the abstract).
Lines 27-29: There are many review articles about the use of agricultural (and wood) waste for composites, authors should mention some of the newest.
Sufficient articles have been added in the Introduction paragraph.
Lines 44-47: Authors should define what they mean by biocomposites.
The essential definition of biocomposites has been added to the Introduction paragraph.
Lines 48-50: Please add also some statistics.
We tried to incorporate some statistics as mentioned in the review, but finally, we decided to leave it “as is” and to provide the complete statistics in the main part of the manuscript. This approach in our opinion will be more profitable for readers.
Lines 54-59: Not only UF, but also PF, MF, MUF, etc.
The sentence has been modified to include other amine resins.
Please discuss more deeply in the Introduction previous research into physical and mechanical properties of bio-based layered hybrid composites.
It is hard to find the previous research to be able to compare to those mentioned in the manuscript, since in our research the biopolymers are not only binders but work also as layers, with a thickness comparable to the wood material thickness. So, finally, we could not find the proper research to compare. Sorry for that.
The relevance should be mentioned in the Introduction.
The information about relevance has been added to the Introduction section.
The results and discussion part are well written.
Thank you!
More discussion with previous research is needed in the discussion part.
We tried to develop the Discussion part by adding some more citations, but, in our opinion, the available data is too far from the scope of the research. We would like to emphasize that in our research biopolymers are not only binders but works also layers, with a thickness comparable to the wood material thickness. So, finally, we could not find the proper research to be able to use in the discussion. Sorry for that.
Please add also limitations of your research, novelty and implications
for further research and industry.
This information has been added to the Conclusion paragraph.
Please find attached file
